# Maximal Update Parametrization and Zero-Shot Hyperparameter Transfer for Fourier Neural Operators

Shanda Li[1]  Shinjae Yoo[2]  Yiming Yang[1]

## Abstract

Fourier Neural Operators (FNOs) offer a principled approach for solving complex partial differential equations (PDEs). However, scaling them to handle more complex PDEs requires increasing the number of Fourier modes, which significantly expands the number of model parameters and makes hyperparameter tuning computationally impractical. To address this, we introduce $\mu$**Transfer-FNO**, a zero-shot hyperparameter transfer technique that enables optimal configurations, tuned on smaller FNOs, to be directly applied to billion-parameter FNOs *without* additional tuning. Building on the Maximal Update Parametrization ($\mu$P) framework, we mathematically derive a parametrization scheme that facilitates the transfer of optimal hyperparameters across models with different numbers of Fourier modes in FNOs, which is validated through extensive experiments on various PDEs. Our empirical study shows that $\mu$Transfer-FNO reduces computational cost for tuning hyperparameters on large FNOs while maintaining or improving accuracy.

## 1. Introduction

With the explosive success of deep learning, Fourier Neural Operators (FNOs) (Li et al., 2021) have emerged as a promising paradigm for solving complex PDEs by learning mappings between function spaces. The kernel integral operator is the most crucial component in FNO. The operator first transforms the input function into the spectral domain and then updates the features for the low-frequency components while zeros out the high-frequency ones. The capacity of FNOs is directly tied to the number of Fourier modes $K$ used in the network. It is desirable to use a large $K$ for modeling complex PDE dynamics and capturing fine-grained spatio-temporal structures.

However, it can be expensive to scale up the number of Fourier modes $K$ in FNO. For a $d$-dimensional PDE, the parameter count of the kernel integral operator scales as $\mathcal{O}(K^d)$, leading to substantial computational requirements. For instance, even for a 4-layer FNO with $d = 3$, increasing $K$ from 3 to 24 expands the model from 1.7M to 906M parameters, with the Fourier integral operator accounting for over 98% of the total parameters.[1] As the parameter count of FNOs grows, hyperparameter tuning—already a computationally expensive process—becomes infeasible at such scales. Standard approaches involve tuning hyperparameters directly on large models, which demands extensive computational resources and is often prohibitive in practice. This bottleneck motivates our central research question:

> *Can we efficiently tune hyperparameters for FNOs with large numbers of Fourier modes?*

One straightforward idea is search for the optimal set of hyperparameters on small FNOs and apply it to large ones. However, naively transferring the hyperparameter across scales can lead to sub-optimal performance. On the left panel of Figure 1, we show that the optimal hyperparameter changes as model scales under the standard training recipe. In this work, we address this challenge by developing a zero-shot hyperparameter transfer technique tailored for FNOs, $\mu$**Transfer-FNO**. The key insight is to properly scale the kernel integral operator parameters as we increase the number of Fourier modes $K$, so that the optimal hyperparameter configuration remain relatively stable across model sizes (e.g., the right panel of Figure 1). This insight suggests that we can identify good hyperparameters using a small proxy model and transfer them to much larger FNOs without additional tuning.

Our approach builds on the Maximal Update Parametrization ($\mu$P) framework (Yang & Hu, 2021). Informally, parametrization refers to the scaling of initialization variances and learning rates of model parameters when the

---

[1]School of Computer Science, Carnegie Mellon University, Pittsburgh, PA, USA [2]Brookhaven National Laboratory, Upton, NY, USA. Correspondence to: Shanda Li `<shandal@cs.cmu.edu>`.

*Proceedings of the 42$^{nd}$ International Conference on Machine Learning*, Vancouver, Canada. PMLR 267, 2025. Copyright 2025 by the author(s).

---

[1]Detailed model configurations can be found in Section 4.1.

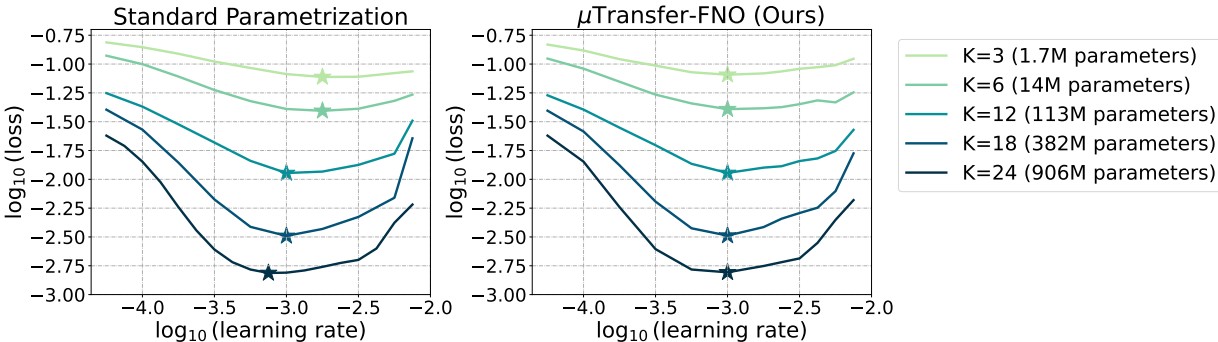

*Figure 1.* **Loss values of FNO-3D on the incompressible Navier-Stokes Equation**, with varying numbers of Fourier modes $K$ and different learning rates. The left and right panels correspond to models trained with standard techniques and $\mu$Transfer-FNO, respectively. The star on each curve marks the optimal learning rate which leads to the lowest loss value. Under standard parametrization, the optimal learning rates shifts as model size increases; while $\mu$Transfer ensures a stable optimal learning rate when the model size scales from 1.7M to near 1B.

model size increases. The Maximal Update Parametrization ($\mu$P) is a unique parametrization with many desirable theoretical and empirical properties which can be rigorously defined and mathematically derived. In particular, prior works show that $\mu$P enables hyperparameter transfer under model width/depth scaling for MLP and Transformers (Yang et al., 2022; 2024). However, whether $\mu$P for neural operators exists and whether it can empirically enable hyperparameter transfer remain unknown. We derive the first $\mu$P formulation for FNO under the scaling of the Fourier mode size $K$. This extension is technically non-trivial due to the unique nature of FNOs: (1) Unlike MLPs and Transformers that process finite-dimensional vectors/sequences, FNOs operate on continuous functions and is agnostic to the discretization; (2) The working mechanism of the kernel integral operator is significantly different from and more complicated than the common deep learning modules such as linear layers or embedding weights analyzed in prior works. Our analysis leads to the following result:

**Theorem 1.1** (Informal version of Theorem 3.5). *The $\mu$P for FNO scales the initialization variances of the kernel integral parameters by $\Theta\left(\frac{1}{d \log K}\right)$ and their learning rates by $\Theta\left(\frac{1}{\sqrt{d \log K}}\right)$, where $d$ and $K$ denotes the PDE dimensionality and number of Fourier modes, respectively.*

As can be seen in the above, the scaling rate is drastically different from all the existing results (e.g., $\Theta(m^{-1})$ for width scaling and $\Theta(L^{-1/2})$ for depth scaling), highlighting the uniqueness of the kernel integral operator and enriching the $\mu$P framework. Technically, this is because existing $\mu$P derivations generally boil down to analyzing the behavior of averages of random variables, while in our deviation, the maximum of $K^d$ (sub-)Gaussian random variables emerges, which naturally leads to a $\sqrt{d \log K}$ term (Vershynin, 2018).

This theoretical result leads to $\mu$Transfer-FNO (Algo-

rithm 1), an efficient hyperparameter tuning strategy for large-scale FNOs. We conduct extensive experiments to validate our theoretical results and demonstrate the effectiveness of $\mu$Transfer. We study three important PDE problems: Burgers' Equation, Darcy Flow, and Navier-Stokes Equation. We empirically show that $\mu$Transfer preserves the optimal learning rate, training batch size, and optimizer configuration across model scales, and can be applied to both the vanilla FNO training and the advanced Physics Informed Neural Operators (Li et al., 2024). Specifically, on Navier-Stokes Equations, $\mu$Transfer-FNO successfully scales to FNOs with nearly 1B parameters and achieves lower test error than the baseline using only $0.30\times$ training compute.

In summary, our contributions in this work are threefold:

- We are the first to derive the Maximal Update Parametrization ($\mu$P) *theory* for FNOs by employing new techniques for analyzing FNOs.

- Based on our derived scaling rates in $\mu$P, we introduce the $\mu$Transfer-FNO *algorithm*, for zero-shot hyperparameter transfer in FNO.

- We *experimentally* validate our theoretical claims on various PDEs, different training algorithms, and different hyper-parameters, showing the robustness of the theory. We also demonstrate that $\mu$Transfer-FNO can significantly reduce computational costs while maintaining or improving accuracy in practice.

## 2. Preliminaries

In this section, we introduce the fundamental concepts of parametric Partial Differential Equations (PDEs), Fourier Neural Operators (FNOs), and $\mu$Transfer.

## 2.1. Parametric Partial Differential Equations

Many problems in science and engineering can be modeled as Partial Differential Equations (PDEs), which describe how physical quantities evolve over time and space. Let $\Omega \subset \mathbb{R}^n$ be a compact domain, $\mathcal{A}$ and $\mathcal{U}$ be the parameter space and solution space which are typically infinite-dimensional function spaces. A general formulation of a parametric PDE is given by:

$$\begin{cases} \mathcal{L}_a u(x) = 0, & x \in \Omega \subset \mathbb{R}^d, \\ \mathcal{B}_a u(x) = 0, & x \in \partial\Omega, \end{cases} \quad (1)$$

where $\mathcal{L}_a$ is a differential operator, $\mathcal{B}_a$ is a boundary condition operator (Kovachki et al., 2023; Saad et al., 2023). Solving the parametric PDE involves finding a **solution operator** $\mathcal{G} : \mathcal{A} \to \mathcal{U}$ whose output $u \in \mathcal{U}$ satisfies both the governing equation and boundary conditions for the given parameter function $a$.

Lu et al. (2021); Kovachki et al. (2023) introduce the idea of learning the solution operator $\mathcal{G}$ with a neural network $\mathcal{G}_\theta$. Given training data $\{(a^{(i)}, u^{(i)})\}_{i=1}^M$, operator learning aims to approximate the solution operator via standard supervised learning.

## 2.2. Fourier Neural Operators

Fourier Neural Operator (FNO) is the most widely used deep learning model for operator learning (Li et al., 2021). Unlike most other neural networks which model mappings between finite-dimensional spaces, FNO operates in infinite-dimensional function spaces as defined below:

$$\mathcal{G}_\theta = \mathcal{Q} \circ \phi\left(\boldsymbol{W}_L + \mathcal{K}_L\right) \circ \cdots \circ \phi\left(\boldsymbol{W}_1 + \mathcal{K}_1\right) \circ \mathcal{P}. \quad (2)$$

In Equation (2), $\mathcal{P}$ and $\mathcal{Q}$ are pointwise linear layers that encode the lower dimension function into higher dimensions and decode the higher dimension function back to the lower dimensions, respectively. $\phi : \mathbb{R} \to \mathbb{R}$ is a point-wise activation function. $\boldsymbol{W}_\ell \in \mathbb{R}^{m \times m}$ is a point-wise linear mapping ($\ell \in [L]$) where $m$ denotes the hidden dimensionality. $\mathcal{K}_\ell$ is the **kernel integral operator** using the Fourier transform, defined as

$$(\mathcal{K}v)(x) = \mathcal{F}^{-1}\left[\boldsymbol{R} \cdot \mathcal{T}_K(\mathcal{F}v)\right](x), \quad \forall x \in \Omega, \quad (3)$$

where $v : \Omega \to \mathbb{R}^m$ is the input function to the operator, $\mathcal{F}$ and $\mathcal{F}^{-1}$ are the $d$-dimensional Fourier transform and its inverse, $\mathcal{T}_K$ truncates the input to the lowest $K$ Fourier modes in each dimension, and $\boldsymbol{R} \in \mathbb{R}^{K^d \times m \times m}$ is the parameter tensor which multiplies with the Fourier feature vector for each of the $K^d$ frequency modes. Typically, $K$ is treated as a tunable hyperparameter of FNO. Using a larger $K$ can make the network more expressive.

In practice, $\Omega$ is typically a rectangular region in $\mathbb{R}^d$ and discretized with $N_1 \times \cdots \times N_d$ points, where $\min_{1 \le j \le d} N_j > K$. The input $v$ is represented as a tensor in $\mathbb{R}^{N_1 \times \cdots \times N_d \times m}$. We assume this in our theoretical analysis.

## 2.3. Maximal Update Parametrization and $\mu$Transfer

Yang & Hu (2021) study the feature learning in over-parametrized networks by considering a rich class of network parametrizations, namely the $abc$-parametrization, and the infinite-width limits. They show that there exists a unique parametrization which enables *maximal* feature learning in a certain sense by scaling the initialization variances and learning rates for model parameters based on theoretical principles. This parametrization scheme is referred as the **Maximal Update Parametrization ($\mu$P)**.

Later, Yang et al. (2022) propose $\mu$Transfer as a novel technique for zero-shot hyperparameter transfer across model scales, which is built upon the $\mu$P. The key insight is that the training dynamics and hence the parameter landscape across remain consistent as the model size scales under $\mu$P. This consistency allows for the optimal hyperparameters identified in a smaller model to be directly applied to a larger target model without additional tuning.

Our work studies $\mu$P and $\mu$Transfer for scaling up the number of Fourier modes $K$ in FNO since it contributes to the major computational cost of the model and is under-explored in existing relevant research. We formalize the terminologies and conduct rigorous derivation in Section 3.

# 3. Theory

In this section, we present our theoretical result that identifies the correct scaling rate for scaling up the kernel integral operator in FNO.

**Notations.** Let $[L] = \{1, \cdots, L\}$. Define the pre-activated and activated hidden features as

$$h_0 = \mathcal{P}v;$$
$$w_\ell = (\boldsymbol{W}_\ell + \mathcal{K}_\ell) h_{\ell-1}, \ h_\ell = \phi(w_\ell) \quad (\ell \in [L]),$$

where $v$ is the input. Further denote by $w_{\ell,t}$ and $h_{\ell,t}$ the features after $t$ training iterations.

## 3.1. Problem setup

We are interested in identifying the Maximal Update Parametrization ($\mu$P) for FNO under the scaling of the number of Fourier modes $K$. The hidden dimensionality $m$ and the model depth $L$ are considered *fixed* in our work, since the scaling of these two factor has been rigorously studied in prior works (Yang & Hu, 2021; Yang et al., 2024).

We extends and generalizes the abc-Parametrization to the parameter tensor $R$ in the kernel integral operator of the Fourier Neural Operator (FNO).

**Definition 3.1** (abc-parametrization for FNO)**.** The generalized abc-parametrization for FNO is specified by mappings $a, b, c : \mathbb{N}^* \to \mathbb{R}_+$ such that

(a) The tensor $\boldsymbol{R}$ in the kernel integral operator (Equation (3)) is parametrized as $\boldsymbol{R} = a(K)\boldsymbol{r}$ for actual trainable parameter $\boldsymbol{r}$ where $K$ denotes the number of truncated Fourier modes;

(b) Each entry of the trainable parameter $\boldsymbol{r}$ is initialized from $\mathcal{N}(0, b(K)^2)$ independently;

(c) The Adam learning rate for optimizing $\boldsymbol{r}$ is scaled as $\eta = c(K)\eta_0$ where $\eta_0$ is the "master" learning rate independent of $K$.

We note that the original definition was restricted to $K^{-s}$ type of scaling, while we will see that the generalization here will indeed facilitate our analysis.

**The standard parametrization for FNO.** Following Yang et al. (2022), we refer to the commonly-used parametrization in the open-source neural operator library[2] as the *standard parametrization*. Under standard parametrization, $a(K) = 1$, $b(K) = m^{-2} = \Theta(1)$, and $c(K) = 1$. As shown in Figure 1, the standard parametrization does not preserve the optimal hyperparameter when using different numbers of Fourier modes, so one cannot reliably transfer the optimal training configuration of small models to large models. More in-depth empirical analysis is presented in Section 4.

Our goal is to find the Maximal Update Parametrization for the kernel integral operator from the parametrization class in Definition 3.1. Maximal Update Parametrization is defined as follows:

**Definition 3.2** (Maximal Update Parametrization for FNO)**.** The Maximal Update Parametrization ($\mu$P) for FNO is an abc-parametrization that satisfies the following properties for any $\ell \in [L]$ with high probability:

• **Stability at initialization**: For a given input function $v$, the hidden features $h_{\ell,0}, w_{\ell,0} = \Theta(1)$.

• **Feature learning in every layer**: For any $t \in \mathbb{N}^*$, the feature update $\Delta_t w_l := w_{l,t} - w_{l,0} = \Theta(1)$.

We point out that the Maximal Update Parametrization is optimizer-dependent: The Maximal Update Parametrization

of MLP is different when using SGD/Adam as the optimizer. In our work, we focus on the Adam optimizer (Kingma & Ba, 2015) because it is the *de facto* choice for training FNO and demonstrates strong empirical performances.

As shown by Yang et al. (2022), $\mu$P of is scale-invariant to the constant shift

$$(a, b, c) \leftarrow \left( \frac{a}{\psi}, b\psi, c\psi \right)$$

for any $\psi : \mathbb{N}^* \to \mathbb{R}_+$ with the Adam optimizer. Therefore, we assume $a(K) \equiv 1$ in our subsequence analysis without the loss of generality. This assumption simplifies the exposition since $\boldsymbol{R}$ itself becomes the actual trainable parameter.

### 3.2. Deriving $\mu$P for FNO

In this subsection, we state our main theoretical result on the $\mu$P for FNO. We begin with formulating the necessary assumptions:

**Assumption 3.3** (Activation functions)**.** We assume the activation function $\phi$ is $\tanh$ or $\sigma$-GELU.

Assumption 3.3 is standard in theoretical works in $\mu$P (Yang & Hu, 2021; Yang et al., 2022; 2024; Ishikawa & Karakida, 2024) and realistic in practice.

**Assumption 3.4** (Sub-Gaussian updates)**.** In the $t$-th iteration, denote by $\boldsymbol{g}_t(\boldsymbol{R}_\ell) \in \mathbb{R}^{K^d \times m \times m}$ the update tensor returned by the optimizer before multiplying with the learning rate. We assume that the entries of $\boldsymbol{g}_t(\boldsymbol{R}_\ell)$ are independent $C$-sub-Gaussian random variables conditioning on the model parameters in the $(t-1)$-th iteration for some constant $C$.

Assumption 3.4 states that the tails of the entries in $\boldsymbol{g}_t(\boldsymbol{R}_\ell)$ decays relatively fast. More discussions can be found in Section 3.3

With the definitions and assumptions stated above, we are ready to state our main theoretical result:

**Theorem 3.5** (Main theoretical result)**.** *Under Assumptions 3.3 and 3.4, the following abc-parametrization is a Maximal Update Parametrization ($\mu$P) of FNO with the Adam optimizer:*

$$a(K) = 1, \ b(K) = c(K) = \Theta\left( \frac{1}{\sqrt{d \log K}} \right), \quad (4)$$

*where $d$ is the PDE dimensionality.*

The detailed proof can be found in Appendix A. The major technique in our proof is to derive the spectral norms of $\mathcal{K}_\ell$ at initialization and its update. Our calculation shows that the spectral norms is the maximum absolute value of $\Theta(K^d)$

(sub-)Gaussian random variables. The scaling functions in Equation (4) control the scales of these variables and hence lead to $\Theta(1)$ spectral norms, which in turn realizes the conditions in Definition 3.2.

Our proof enriches the technique for $\mu$P deviations. In prior works (Yang & Hu, 2021; Yang et al., 2022), $\mu$P deviations mainly rely on the central limit theorem and the law of large numbers to characterize the *average* of random variables. By contrast, in our study, the unique structure of FNO and the kernel integral operator leads to the analysis of the *maximum* of random variables and consequently an interesting scaling function.

We also note that the scaling function does not depend on $N_1, \cdots, N_j$, i.e., the discretization of the function domain $\Omega$. This property is appealing in practice because it makes sure that $\mu$P does not change the resolution-agnostic nature of FNOs.

In the subsequent sections, we show that the theoretical result is practically useful by presenting the $\mu$Transfer-FNO algorithm and empirical study.

### 3.3. Discussions on the result

In this section, we make a few remarks on the theoretical finding.

**Regarding the connection to Transformers.** Recent research has shown that the Transformer attention module and its variants (e.g., Continuum Attention) are closely related to operator learning (Kovachki et al., 2023; Rahman et al., 2024; Calvello et al., 2024). While the Maximal Update Parametrization ($\mu$P) for FNO has been established by Yang et al. (2022), we note that the result is not readily applicable to the Fourier Integral Operator.

Specifically, for attention module and its variants (e.g., Continuum Attention), the parametrization closely resembles vanilla attention, and its size is controlled by the hidden dimensionality (i.e., the model "width"). Hence existing results on scaling up model "width" are applicable. In contrast, the Fourier Integral Operator is parametrized in a uniquely different way – by modeling the Fourier Transform of a kernel function with $K$ Fourier modes. This design differs from all modules which have been explicitly studied in existing literature. Theorem 3.5 shows that its scaling rate is also different from that of other model components. That being said, in Fourier Integral Operators, the notion of model "width" still exists (corresponding to $m$ in Section 2), and existing width scaling results may apply to it.

**Regarding Assumption 3.4 and proactical applications of Theorem 3.5.** Assumption 3.4 requires sub-Gaussian updates to the parameter tensor $\boldsymbol{R}$ in the kernel integral operator. While this is not standard in existing analysis, it is natural for the Adam optimizer where entry-wise normalized gradient momentum is used in the update and the normalization controls the scale of the update. Furthermore, the sub-Gaussian condition can be explicitly enforced in practice through *element-wise gradient clipping* (Pascanu et al., 2013), which bounds the updates and thereby ensures their sub-Gaussianity.

### 3.4. The $\mu$Transfer-FNO algorithm

As shown by Yang et al. (2022), the hyperparameter landscape across neural networks of different sizes (e.g., FNOs with different numbers of Fourier modes) is reasonably stable if parametrized according to $\mu$P. This property enables us to probe the hyperparameter landscape with a small fixed-sized proxy model and identify the optimal configuration. The optimal configuration will also be (near) optimal for the large model under $\mu$P, thus circumventing the direct and computationally intensive hyperparameter tuning of large models. We present this procedure in Algorithm 1.

---

**Algorithm 1** $\mu$Transfer-FNO

---

**Require:** Hyper-parameter search space $\Xi$.
    FNO training algorithm $\texttt{train}(\xi, K)$ which returns the final loss given hyperparameter $\xi$ and number of Fourier modes $K$.
    Numbers of Fourier modes for target and small proxy models $K^*$ and $K_{\text{proxy}}$.
**Ensure:** Efficiently train a target FNO using (near) optimal configurations.

1: $\triangleright$ PARAMETER SWEEPING ON SMALL MODELS.
2: $\xi^* \leftarrow \operatorname{argmin}_{\xi \in \Xi} \texttt{train}(\xi, K_{\text{proxy}})$.

3: $\triangleright$ PARAMETER RESCALING.

4: $\xi^*_{\text{learning rate for } \boldsymbol{R}} \leftarrow \sqrt{\dfrac{\log K_{\text{proxy}}}{\log K^*}} \xi^*_{\text{learning rate for } \boldsymbol{R}}$

5: $\xi^*_{\text{init. var. of } \boldsymbol{R}} \leftarrow \dfrac{\log K_{\text{proxy}}}{\log K^*} \xi^*_{\text{init. var. of } \boldsymbol{R}}$

6: $\triangleright$ TARGET MODEL TRAINING.
7: $\texttt{train}(\xi^*, K^*)$

---

## 4. Experiments

In this section, we empirically validate our theory and study the effectiveness of the generality, efficiency, and effectiveness $\mu$Transfer-FNO algorithm (Algorithm 1). In particular, we aim at answering the following questions through experiments:

- Does the $\mu$P for FNO stated in Theorem 3.5 preserve

the optimal learning rate?

- Is the $\mu$Transfer-FNO algorithm generalizable to more advanced FNO training techniques, e.g., Physics Informed Neural Operator (PINO) (Li et al., 2024)?

- Is the $\mu$Transfer-FNO algorithm applicable to transfer hyperparameters across model scales beyond the learning rate?

- How much computation cost does the $\mu$Transfer-FNO algorithm save compare to the naive hyperparameter tuning strategy?

We first detail the problem setup and training configurations in our empirical study, and then presents experimental results to answer the above research questions in the subsequent subsections. We release our code at https://github.com/LithiumDA/muTransfer-FNO.

### 4.1. Problem setup and training configurations

**Burgers' Equation.** We use the 1D Burgers' Equation as the testbed for the FNO-1D model. The equation takes the following form:

$$\begin{cases} \partial_t u(x,t) + \partial_x(u^2(x,t)/2) = \nu \partial_{xx} u(x,t) & t \in (0,1] \\ u(x,0) = u_0(x) & x \in (0,1) \end{cases}$$

where $u_0$ is the initial condition and $\nu$ is the viscosity coefficient. We assume periodic boundary conditions. In the above equation, $x$ is the spatial variable, while $t$ is the temporal variable, which is slightly different from the notation in Equation (1) where $x$ is denotes as the spatiotemporal variable in the PDE. We hope this abuse of notations does not confuse the readers.

We aim to learn the mapping from the initial condition to the solution, $\mathcal{G} : u_0(\cdot) \mapsto u(\cdot, 1)$. The spatial domain is discretized into 8192 grids and downsampled to 1024 during training. The viscosity coefficient is set to $0.1$. We train FNO-1D with 4 layers, 64 hidden dimensions, the GELU activation function, and varying numbers of Fourier modes $K$. We use $800/200$ samples for training/evaluation. The model is trained with a batch size of 20 for 750 epochs, and the learning rate halves every 50 epochs. We use Adam (Kingma & Ba, 2015) as the optimizer and set its $(\beta_1, \beta_2)$ to $(0.9, 0.999)$ in this and all the subsequent experiments, unless otherwise specified.

**Darcy Flow.** We test FNO-2D models on the steady-state of the 2D Darcy Flow on the unit square. It is a second-order linear elliptic PDE:

$$\begin{cases} -\nabla \cdot (a(x)\nabla u(x)) = 1 & x \in (0,1)^2 \\ u(x) = 0 & x \in \partial(0,1)^2 \end{cases}.$$

We train FNO-2D to learn the mapping from the PDE co-efficient to the solution, $\mathcal{G} : a \mapsto u$. The data resolution is $421 \times 421$ and downsampled $61 \times 61$ during training. We train FNO-2D with 4 layers, 64 hidden dimensions, the GELU activation function, and varying numbers of Fourier modes $K$. We use $1000/500$ samples for training/evaluation. The model is trained with a batch size of 20 for 300 epochs. The learning rate halves at the 100th, 150th, and 200th epoch by default. We also explore other batch sizes and optimizer hyperparameters in Section 4.4.

**Navier-Stokes Equation.** We evaluate FNO-3D on the incompressible Navier-Stokes equations for a viscous, in-compressible fluid in vorticity form on a torus in the 2D space:

$$\begin{cases} \partial_t \omega + u \cdot \nabla \omega = \frac{1}{\text{Re}} \Delta \omega + f & x \in [0, 2\pi]^2, t \in (0, T] \\ \nabla \cdot u(x,t) = 0 & x \in [0, 2\pi]^2, t \in [0, T] \\ \omega(x,0) = \omega_0(x) & x \in [0, 2\pi]^2 \end{cases}.$$

In this equation, $\omega$ is the vorticity field, $u$ is the velocity field, $\text{Re}$ is the Reynolds number characterizing fluid viscosity, and $f$ is the forcing term. The equations use periodic boundary conditions. The data is generated on $512 \times 512$ spatial grids and downsampled to $64 \times 64$ resolution. The temporal resolution is 512 steps per unit time, and we set the time interval $[0, T]$ to $[0, 0.125]$. We set the Reynolds number $\text{Re}$ to 500 and the forcing term $f$ to $-4\cos(4x_2)$.

We train FNO-3D to do spatial-temporal convolution jointly following Li et al. (2021; 2024); George et al. (2024). The model contains 4 layers, 64 hidden dimensions, the GELU activation function, and varying numbers of Fourier modes $K$. We use $800/200$ samples for training/evaluation. The model is trained with a batch size of 2 for 50000 iterations. The learning rate halves at the 20000th and 40000th iteration. For $\mu$Transfer-FNO, we apply element-wise gradient clipping to the kernel integral operator module and the clip value is set to $0.01$.

The experiments for FNO-1D and FNO-2D are run on a NVIDIA GeForce RTX 2080 Ti GPU with 11GB memory. The experiments for FNO-3D are run on a NVIDIA RTX A6000 GPU with 50GB memory. Our code is built upon the open source library for Physics-informed Neural Operator[3] using PyTorch (Paszke et al., 2019).

### 4.2. Identifying optimal learning rates under scaling

We first examine how the learning rate landscape changes as we scale up the number of Fourier modes $K$ on the three PDE problems. Figures 1, 2a and 2b show the loss values

---

[3]https://github.com/neuraloperator/physics_informed

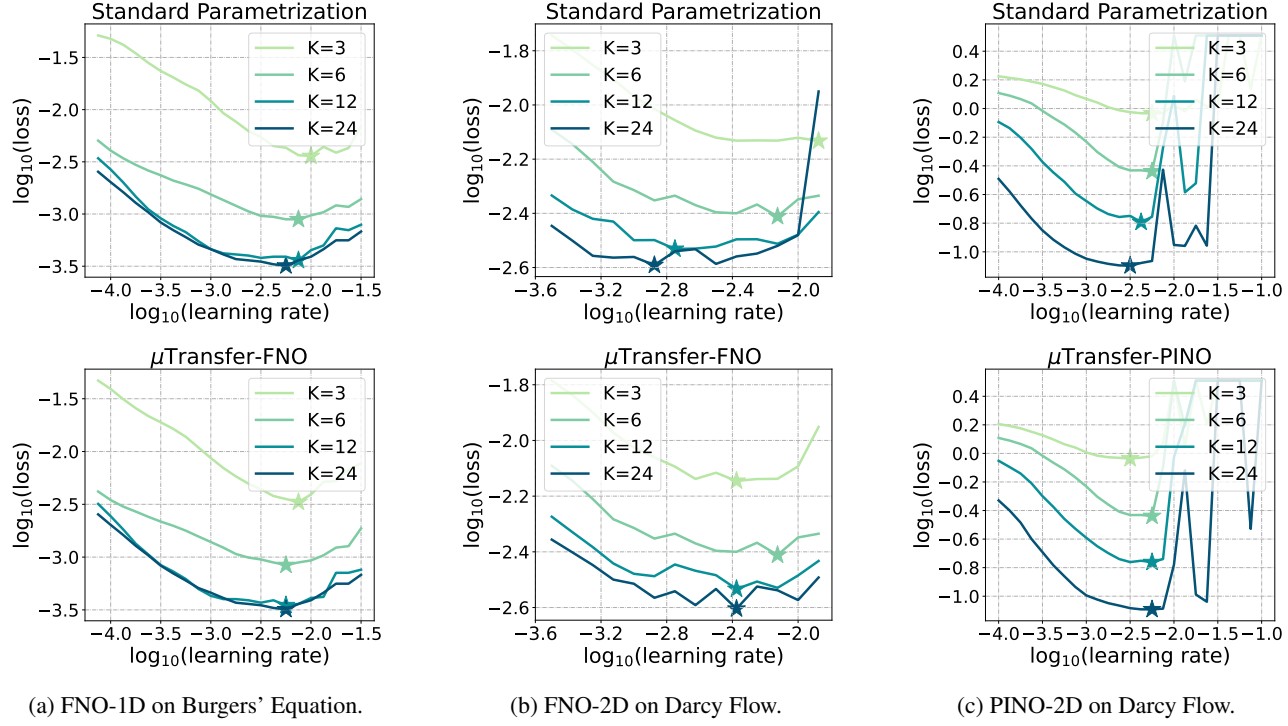

(a) FNO-1D on Burgers' Equation.     (b) FNO-2D on Darcy Flow.     (c) PINO-2D on Darcy Flow.

*Figure 2.* **Loss values of FNOs/PINOs with varying numbers of Fourier modes $K$ and different learning rates**. The star on each curve marks the optimal learning rate which leads to the lowest loss value. We observe that the optimal learning rate is more consistent under $\mu$Transfer, compared to the standard parametrization.

under both standard parametrization and $\mu$Transfer-FNO trained with different learning rates.

Under standard parametrization (the left panel in Figure 1 and the top row in Figure 2), we consistently observe a clear shift in the optimal learning rate as $K$ increases. For example, for FNO-3D on the the incompressible Navier-Stokes Equation, the optimal learning rate (marked by stars) shifts from $1.8 \times 10^{-3}$ when $K = 3$ to $7.4 \times 10^{-4}$ when $K = 24$. This shift indicates that naively transferring hyperparameters tuned on small models to larger ones leads to suboptimal training configurations.

In contrast, when using $\mu$Transfer-FNO (the right panel in Figure 1 and the bottom row in Figure 2), the optimal learning rates remain remarkably stable across different values of $K$. For example, for FNO-2D on the the incompressible Navier-Stokes Equation, the optimal learning rate align approximately at $4.2 \times 10^{-3}$. This stability validates our argument in Theorem 3.5 and Section 3.4. That being said, the optimal learning rate for a certain model size may sometimes deviate from the expected value, e.g., on the curves of $K = 6$ in Figure 1 (right), $K = 3$ in Figure 2a (bottom), and $K = 6$ in Figure 2b (bottom). We note that this instability typically occurs when $K$ is very small and attribute this to the randomness during training.

Furthermore, we observe that under $\mu$Transfer-FNO, the loss curves exhibit more consistent shapes across different $K$ values, suggesting that the overall optimization landscape becomes more uniform.

### 4.3. Applications to Physics-Informed Neural Operators

All the results presented in Section 4.2 are based on standard supervised training of FNO. In this subsection, we further evaluate the effectiveness of our approach on Physics-Informed Neural Operators (PINO) which incorporates PDE constraints into the training objective (Li et al., 2024). Note that PINO does not change the architectural design of FNO. Hence the parametrization scheme developed in Theorem 3.5 is still valid. We refer to our approach in this setting as $\mu$Transfer-PINO. We use the same training recipe as described in Section 4.2, and set the ratio between the data loss and physics-informed loss to $5 : 1$.

We present experimental results of the standard parametrization and $\mu$Transfer-PINO on the Darcy flow problem in Figure 2c. The results share similar trends with those in Section 4.2. Under standard parametrization, PINO exhibits sensitivity to the learning rate as $K$ increases. The optimal learning rate gradually decreases when $K$ increase. In contrast, $\mu$Transfer successfully stabilizes the learning rate landscape across different $K$ values for PINO, with the op-

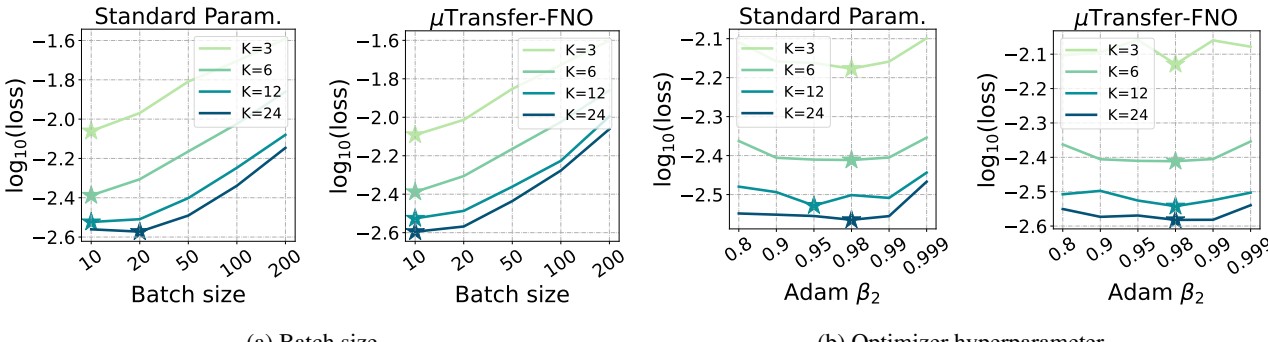

*(a) Batch size.*    *(b) Optimizer hyperparameter.*

*Figure 3.* **Loss values of FNO-2D on the Darcy Flow problem with different hyperparameters** and varying numbers of Fourier modes $K$. The star on each curve marks the optimal setting which leads to the lowest loss value. We observe that the optimal hyperparameter is more consistent under $\mu$Transfer, compared to the standard parametrization.

timal learning rates aligning at approximately $5.6 \times 10^{-3}$. This result is noteworthy given that PINO's training dynamics are typically more complex due to the additional physics-informed loss terms. The consistent performance across both standard FNO and PINO shows the generality of Theorem 3.5, independent of the specific training objective variations.

### 4.4. Applications to more hyperparameters

The experiments in Sections 4.2 and 4.3 all focus on the learning rate. In this subsection, we investigate whether $\mu$Transfer-FNO's benefits extend to other hyperparameters beyond learning rates. Specifically, we consider tuning the batch size and Adam's $\beta_2$ parameter for FNO-2D on the Darcy flow problem.

Our experimental results are shown in Figure 3. For batch size (Figure 3a), standard parametrization exhibits varying optimal values across different $K$: larger models ($K = 24$) require a larger batch size, while smaller models perform better with smaller batches. In contrast, under $\mu$Transfer-FNO, the optimal batch size is consistently 20 regardless of model size. Similarly, for Adam's $\beta_2$ parameter (Figure 3b), standard parametrization shows shifting optimal values as $K$ increases. With $\mu$Transfer-FNO, the optimal $\beta_2$ aligns at 0.98 across all model sizes. This consistency further validates that $\mu$Transfer-FNO stabilizes the overall optimization dynamics. These results demonstrate that $\mu$Transfer-FNO enables reliable hyperparameter transfer across model scales for multiple optimization-related hyperparameters, making it a practical tool for efficient large-scale FNO training.

### 4.5. Practical effectiveness of $\mu$Transfer-FNO

To demonstrate the effectiveness of $\mu$Transfer-FNO in practice, we compare model performances of directly tuning the large model and applying the $\mu$Transfer-FNO algorithm. The results are reported in Table 1. First, we note that

$\mu$Transfer-FNO leads to lower $L^2$ relative test error on the test set. This is natural because $\mu$Transfer-FNO sweeps hyperparameters on small models, hence can use a larger hyperparameter search space. Second, $\mu$Transfer-FNO surpasses the test performance of the naive hyperparameter tuning method on the Navier-Stokes Equation while only uses $0.30\times$ of the total training compute, which clearly demonstrates the benefit of tuning small proxy models to identify the hyperparameter choice. Finally, we note that $\mu$Transfer-FNO is slower than the naive approach on the Darcy Flow problem because the training cost gap between small models and large models is not extremely large in this setting. In practice, $\mu$Transfer-FNO holds stronger potential in the large-scale FNO training settings, e.g., when scaling to billion-parameter sizes.

## 5. Related Works

### 5.1. Neural PDE solvers

with the growth of available data and computational resources, there have been growing interests in developing neural network approaches to partial differential equation solving (Han et al., 2018; Khoo et al., 2021; Kochkov et al., 2021; Wang et al., 2022). One stream of works leverages neural networks as a new ansatz of solutions, notable examples include (Sirignano & Spiliopoulos, 2018; Raissi et al., 2019; Zang et al., 2020; He et al., 2023). Another active research direction is operator learning, where neural networks serve as an ansatz of solution operator (Long et al., 2018; 2019; Lu et al., 2021; Li et al., 2021; Cao, 2021; Kovachki et al., 2023; Wang et al., 2024). A few works present approaches to combining the strengths of these two paradigms (Huang et al., 2022; Li et al., 2024).

Our work falls into the operator learning category and specifically focuses on tuning large-scale FNO. There are several prior works studying a similar topic: George et al. (2024) introduce Incremental Fourier Neural Operator (iFNO) which

*Table 1.* **Test relative error comparisons** between directly tuning the large model and $\mu$Transfer-FNOs.

| Model | FNO-2D | | FNO-3D | |
|---|---|---|---|---|
| Problem | Darcy Flow | | Navier-Stokes Equation | |
| Metric | $L^2$ relative error | Total training cost | $L^2$ relative error | Total training cost |
| Tune the full model | 1.25% | $\mathbf{1}\times$ | 5.69% | $1\times$ |
| $\mu$Transfer-FNO (ours) | **1.22**% | $1.38\times$ | **5.34**% | $\mathbf{0.30}\times$ |

progressively increases both the number of Fourier modes and the training data resolution to enhance training efficiency. Alkin et al. (2024) propose Universal Physics Transformers (UPTs) as an efficient and unified approach to a wide range of spatio-temporal problems by propagating dynamics in the latent space. While this paper presents improved model architecture for operator learning, our work studies efficient hyperparameter searching for the most commonly-used FNO architecture.

### 5.2. Theoretical analysis on FNOs

Theoretical analysis of FNOs provides important research insights into model designs and applications. Kovachki et al. (2021) prove the universal approximation theorem for FNOs. De Ryck & Mishra (2022) prove the approximation error bounds on physics-informed operator learning. Koshizuka et al. (2024) delve into the expressivity and trainability of FNOs using mean-field theory. Kim & Kang (2024) study the generalization capabilities of FNOs by analyzing their Rademacher complexity. Le & Dik (2024) systematically present a series of mathematical analysis of neural operator. Our work formulates and derives the Maximal Update Parametrization of FNO and presents the application to zero-shot hyperparameter transfer.

### 5.3. $\mu$P, $\mu$Transfer, and Scaling in deep learning

Our work builds upon the $\mu$P (Yang & Hu, 2021) and $\mu$Transfer (Yang et al., 2022) framework. The framework is initially introduced for analyzing the limits under model width scaling and gradient descent training, and later extended to depth scaling (Yang et al., 2024; Bordelon et al., 2024; Chen et al., 2024) and second-order optimization (Ishikawa & Karakida, 2024). Dey et al. (2023); Lingle (2024) present empirical applications of $\mu$Transfer on Transformer-based large language model training. Blake et al. (2024) incorporate Unit Scaling into $\mu$P, combining the advantages of hyperparameter transfer and stable training in FP8 precision. Noci et al. (2024) show some properties of the loss landscape are preserved across model sizes under the right scaling of the architecture. Our work is, to the best of our knowledge, the first analysis on $\mu$P and $\mu$Transfer for PDE solving and scaling up the FNO architecture.

More broadly, scaling up deep learning models is currently an active research topic. Existing research has derived prin-

cipled scaling laws and demonstrated impressive empirical success across a wide range of domains, including board games (Jones, 2021), language modeling (Kaplan et al., 2020; Hoffmann et al., 2022; OpenAI, 2023), image modeling (Henighan et al., 2020; Yu et al., 2022; Peebles & Xie, 2023), mathematical reasoning (Snell et al., 2024; Wu et al., 2024), and very recently, PDE solving (Fan et al., 2025; Li et al., 2025). We believe this direction holds strong promise in neural PDE solvers and that our work contributes to this emerging area.

## 6. Conclusions and Limitations

**Conclusions.** We introduce $\mu$Transfer-FNO, a principled method for zero-shot hyperparameter transfer in FNO. We derive the $\mu$P for FNO theoretically, and our proof offers new technical insights to the study of $\mu$P. On a wide range of PDEs, we experimentally demonstrate that the parametrization scheme preserves the optimal hyperparameters under various setting. Our results show that $\mu$Transfer-FNO is theoretically-principled and can serve as a practical and efficient strategy for tuning neural PDE solvers in billion-parameter scales.

**Limitations.** Although FNO is widely used in neural network methods for PDE solving, there are many other neural operator variants including Multipole Graph Neural Operator (Li et al., 2020), DeepONets (Lu et al., 2021), Multi-wavelet NO (Gupta et al., 2021), and Transformer-based foundation models (Ye et al., 2024). Our results are dedicated to FNO and extending the results to more general neural operator variants can be avenues for future research.

## Acknowledgements

The authors would like to thank the constructive suggestions from Chuwei Wang (Caltech) and the anonymous reviewers.

This material is based upon work supported in part by the U.S. Department of Energy, Office of Science, Office of Advanced Scientific Computing Research under Contract DE-AC02-06CH11357. This research uses resources of the Oak Ridge Leadership Computing Facility and NERSC, which are a DOE Office of Science User Facility supported under Contract DE-AC05-00OR22725 and under Contract No. DE-AC02-05CH11231 using NERSC award ALCC-ERCAP0034238.

## Impact Statement

This paper presents work whose goal is to advance the field of Machine Learning. There are many potential societal consequences of our work, none which we feel must be specifically highlighted here.

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

# A. Omitted theoretical results

## A.1. Technical lemmas

We first present a few technical lemma as preparations for the main proof.

**Lemma A.1.** *For any $\boldsymbol{R}$, $\mathcal{K}$ is a linear operator.*

*Proof.* We note that the Fourier transform, the truncation operator, the multiplication with $\boldsymbol{R}$, and the inverse Fourier transform are all linear operators. $\mathcal{K}$, as the composition of these linear operators, is still a linear operator. $\square$

To highlight the kernel integral's dependency on the parameter tensor $\boldsymbol{R}$, we slightly abuse notation and write $\mathcal{K} = \mathcal{K}^{\boldsymbol{R}}$.

**Lemma A.2.** $\mathcal{K}^{\boldsymbol{R}}$ *is linear in* $\boldsymbol{R}$.

*Proof.* The proof is straightforward based on the definition in Equation (3) and the linearty of the inverse Fourier transform. $\square$

**Lemma A.3** (Feature learning condition). *A parametrization of FNO satisfies feature learning in every layer if and only if the followings are both true with high probability*

- *The feature coordinate size $h_{\ell,t}$ remain stable, i.e., $h_{\ell,t} = \Theta(1)$ for every $\ell \in \{0\} \cup [L]$ and $t \in \mathbb{N}$.*

- *$\mathcal{K}_\ell$ updates maximally for every $\ell \in [L]$ and $t \in \mathbb{N}^*$, i.e., $\Delta_t \mathcal{K}_\ell h_{\ell-1,t} = \Theta(1)$ where $\Delta_t \mathcal{K}_\ell = \mathcal{K}_{\ell,t} - \mathcal{K}_{\ell,0}$.*

*Proof.* We note that $\mathcal{P}$, $\boldsymbol{W}_1 + \mathcal{K}_1$, $\cdots$, $\boldsymbol{W}_L + \mathcal{K}_L$, and $\mathcal{Q}$ are all linear operators over the input functions, i.e., linear mappings over the input tensor in practice where functions are discretized. Therefore, the feature learning conditions (Conditions A.1 & A.2 in (Ishikawa & Karakida, 2024)) still hold in our setting.

According the conditions, FNO satisfies feature learning in every layer if and only if the following two conditions are satisfied with high probability:

- **Maximal updates:** $\Delta_t(\boldsymbol{W}_\ell + \mathcal{K}_\ell)h_{\ell-1,t} = \Theta(1)$;

- **Maximal initialization:** $\mathcal{Q}_0 \Delta_t h_L = \Theta(1)$.

For the first condition, It is easy to note that $\Delta_t \boldsymbol{W}_\ell$ does not scale when $K$ increases. Thus, $\Delta_t(\boldsymbol{W}_\ell + \mathcal{K}_\ell)h_{\ell-1,t} = \Theta(1)$ is equivalent to $\Delta_t \mathcal{K}_\ell h_{\ell-1,t} = \Theta(1)$, which we assume to be true (with high probability) in the lemma statement.

For the second condition is easy to check because the feature coordinate size $h_{L,0}, h_{L,t} = \Theta(1)$. Thus $\Delta_t h_L = \Theta(1)$. Also, the spectral norm of $\mathcal{Q}_0$ is $\Theta(1)$ and does not scale with $K$. Thus $\mathcal{Q}_0 \Delta_t h_L = \Theta(1)$ in the general case.

Therefore, the assumptions in the lemma statement implies Conditions A.1 & A.2 in (Ishikawa & Karakida, 2024), and hence implies feature learning. $\square$

## A.2. Proof of Theorem 3.5

In this subsection, we present the complete proof of the main theoretical result, Theorem 3.5.

*Proof.* We are only interested in scaling up $K$ with $m$ fixed. Thus, without the loss of generality, we assume $m = 1$. The proof can easily be extended to the case with a general $m$.

In the kernel integral operator $\mathcal{K}_\ell$, let the parameter $\boldsymbol{R}_\ell = \left\{r_\ell^{\boldsymbol{k}}\right\}_{\boldsymbol{k} \in [K]^d}$ where $r_\ell^{\boldsymbol{k}} \in \mathbb{R}$.

**Stability at initialization.** We first consider the model behavior at initialization and prove the stability condition by induction. Specifically, given $h_0 = \Theta(1)$ at initialization, it suffices to show that for $\ell \in [L]$, $w_{\ell,0} = \Theta(1)$ and $h_{\ell,0} = \Theta(1)$ if $h_{\ell-1,0} = \Theta(1)$. For ease of notations, we omit the subscript 0 in the subsequent analysis on the initialization.

By definition, $w_\ell = \boldsymbol{W}_\ell h_{\ell-1} + \mathcal{K}_\ell h_{\ell-1}$. Obviously, $\boldsymbol{W}_{\ell-1} = \Theta(1)$.

Regarding $\mathcal{K}_\ell h_\ell$, Lemma A.1 implies that $\mathcal{K}_t$ is a linear operator. Given the discretized input $h_\ell \in \mathbb{R}^{N_1 \times \cdots \times N_d}$, $\mathcal{K}_\ell$ can be written as a matrix as follows:

$$\boldsymbol{K}_\ell = \boldsymbol{F}^{\mathsf{H}} \widetilde{\boldsymbol{R}}_\ell \boldsymbol{F} \tag{5}$$

where $\boldsymbol{F}$ denotes the $N_1 \times \cdots \times N_d$ (multidimensional) discrete Fourier transform matrix, the superscript H denotes the conjugate transpose, and $\widetilde{\boldsymbol{R}}_\ell \in \mathbb{R}^{N_1 \cdots N_d \times N_1 \cdots N_d}$ accounts for the truncation operator and multiplication with $\boldsymbol{R}_\ell$. The rows and columns are indexed by $\boldsymbol{j} \in [N_1] \times \cdots \times [N_d]$. Specifically, its $(\boldsymbol{j}, \boldsymbol{k})$-th entry is

$$\widetilde{R}_\ell^{(\boldsymbol{j},\boldsymbol{k})} = \begin{cases} r_\ell^{\boldsymbol{k}} & \boldsymbol{j} = \boldsymbol{k} \in [K]^d \\ 0 & \text{Otherwise} \end{cases}.$$

Note that $\boldsymbol{F}$ is an orthogonal matrix and $\widetilde{\boldsymbol{R}}$ is a diagonal matrix, hence Equation (5) is the eigen-decomposition of $\boldsymbol{K}_\ell$. Therefore, the spectral norm of $\boldsymbol{K}_\ell$ is

$$\|\boldsymbol{K}_\ell\|_2 = \max_{\boldsymbol{k} \in [K]^d} \left| r_\ell^{\boldsymbol{k}} \right|.$$

Recall that $\{r_\ell^{\boldsymbol{k}}\}_{\boldsymbol{k} \in [K]^d}$ are $K^d$ i.i.d. random variables sampled from $\mathcal{N}(0, b(K)^2)$. By standard result in high-dimensional probability (Vershynin, 2018), with high probability,

$$\|\boldsymbol{K}_\ell\|_2 = \max_{\boldsymbol{k} \in [K]^d} \left| r_\ell^{\boldsymbol{k}} \right| = \Theta\left( b(K)\sqrt{d \log K} \right) = \Theta(1).$$

Also note that $h_{\ell-1} = \Theta(1)$. Thus, $\mathcal{K}_\ell h_{\ell-1} = \Theta(1)$ and hence $w_\ell = \boldsymbol{W}_\ell h_{\ell-1} + \mathcal{K}_\ell h_{\ell-1} = \Theta(1)$. Finally, by Assumption 3.3, $h_\ell = \phi(w_\ell) = \Theta(1)$. Therefore, we show the stability at initialization by induction on $\ell$.

**Feature learning in every layer.** We begin with characterizing the updates of $\mathcal{K}_{\ell,t}$ in each iteration $t$. For $t \in \mathbb{N}$, by Lemma A.2,

$$\mathcal{K}_{\ell,t+1} - \mathcal{K}_{\ell,t} = \mathcal{K}^{\boldsymbol{R}_{\ell,t+1}} - \mathcal{K}^{\boldsymbol{R}_{\ell,t}} = \mathcal{K}^{\boldsymbol{R}_{\ell,t+1} - \boldsymbol{R}_{\ell,t}} = \eta_0 c(K) \mathcal{K}^{\boldsymbol{g}_t(\boldsymbol{R}_\ell)},$$

where $\boldsymbol{g}_t(\boldsymbol{R}_\ell)$ is defined in Assumption 3.4. By Assumption 3.4, using an argument similar to the above proof of stability at initialization, we have

$$\|\boldsymbol{K}_{\ell,t+1} - \boldsymbol{K}_{\ell,t}\|_2 = \Theta(1),$$

where $\boldsymbol{K}_{\ell,t}$ denotes the discretized operator $\mathcal{K}_{\ell,t}$.

Now we prove feature learning in every layer by verifying the conditions in Lemma A.3:

- *The feature coordinate size $h_{\ell,t}$ remain stable, i.e., $h_{\ell,t} = \Theta(1)$ for every $\ell \in \{0\} \cup [L]$ and $t \in \mathbb{N}$.*

  We prove by induction on $\ell$.

  First, it is easy to see the $h_{0,t} = \Theta(1)$ for $t \in \mathbb{N}$ because $\mathcal{P}$ does not scale with $K$.

  Second, assume that $h_{\ell-1,t} = \Theta(1)$ for some $\ell \in [L]$. Then

  $$\|\boldsymbol{K}_{\ell,t}\|_2 \leq \|\boldsymbol{K}_{\ell,0}\| + \sum_{\tau=0}^{t-1} \|\boldsymbol{K}_{\ell,\tau+1} - \boldsymbol{K}_{\ell,\tau}\|_2 = \Theta(1) \quad \Rightarrow \quad \|\boldsymbol{K}_{\ell,t} h_{\ell-1,t}\| = \Theta(1).$$

Also note that $\boldsymbol{W}_{\ell,t}$ does not scale with $K$. Thus,

$$w_{\ell,t} = (\boldsymbol{W}_{\ell,t} + \boldsymbol{K}_{\ell,t})h_{\ell-1,t} = \Theta(1); \ h_{\ell,t} = \phi(w_{\ell,t}) = \Theta(1).$$

Therefore, $h_{\ell,t} = \Theta(1)$ for every $\ell \in \{0\} \cup [L]$ and $t \in \mathbb{N}$.

- $\mathcal{K}_\ell$ *updates maximally for every* $\ell \in [L]$ *and* $t \in \mathbb{N}^*$, *i.e.,* $\Delta_t \mathcal{K}_\ell h_{\ell-1,t} = \Theta(1)$ *where* $\Delta_t \mathcal{K}_\ell = \mathcal{K}_{\ell,t} - \mathcal{K}_{\ell,0}$.
  In the above we have shown that $h_{\ell-1,t} = \Theta(1)$. Moreover,

$$\|\Delta_t \boldsymbol{K}_\ell\|_2 = \|\boldsymbol{K}_{\ell,t} - \boldsymbol{K}_{\ell,0}\|_2 \leq \sum_{\tau=0}^{t-1} \|\boldsymbol{K}_{\ell,\tau+1} - \boldsymbol{K}_{\ell,\tau}\|_2 = \Theta(1).$$

  Therefore, in the general case, $\Delta_t \mathcal{K}_\ell h_{\ell-1,t} = \Theta(1)$, which concludes the proof.

Finally, according to Lemma A.3 and the above two statement, we prove feature learning in every layer.

By Definition 3.2, the parametrization in Theorem 3.5 is $\mu$P for FNO. □

