# OpenReview forum: "Maximal Update Parametrization and Zero-Shot Hyperparameter Transfer for Fourier Neural Operators"
_ICML.cc/2025/Conference — ICML 2025 poster_

### Official Review · Reviewer_r9fK · 2025-03-06

**Overall Recommendation:** 2

**Summary:**

This paper applies the Maximum Update Parametrization (µP) framework to Fourier Neural Operators (FNO), demonstrating that a single set of hyperparameters can effectively work for both large-scale and small-scale FNO models.

**Claims And Evidence:**

While I understand the authors' claims, the significance of this research question remains unclear. Scaling up transformer-like models appears to be a more straightforward alternative.

**Essential References Not Discussed:**

N/A

**Experimental Designs Or Analyses:**

The datasets used in this study appear overly simplistic. The necessity for FNO models with billions of parameters in these tasks is questionable. More challenging datasets, such as those involving multiple mixed equations for training PDE foundation models [1,2], would provide more meaningful scenarios for investigating hyperparameter tuning in large-scale models.
[1] Unisolver: PDE-Conditional Transformers Are Universal PDE Solvers
[2] PDEformer: Towards a Foundation Model for One-Dimensional Partial Differential Equations

**Methods And Evaluation Criteria:**

The authors apply a framework for transferring hyperparameters from small to large FNO models.

**Other Comments Or Suggestions:**

N/A

**Other Strengths And Weaknesses:**

The rationale for scaling up FNO is unclear, as FNO itself does not seem particularly suitable for scaling. Current transformer-based models demonstrate superior performance and stronger scaling capabilities compared to FNO, raising questions about the significance of scaling up FNO.

**Questions For Authors:**

N/A

**Relation To Broader Scientific Literature:**

This work may contribute to hyperparameter tuning research.

**Theoretical Claims:**

I have not verified the details of the theoretical proofs.

---

> ### Author Rebuttal · Authors · 2025-04-01
>
> We sincerely appreciate your thoughtful comments and constructive suggestions. Let us respond to your concerns one-by-one below.
>
> **Regarding the significance of our work.**
>
> Our main contributions can be summarized as follows:
>
> * **On the theory side:** We are the first to derive the Maximum Update Parametrization ($\mu$P) for FNOs, identifying the unique scaling rate for kernel integral parameters that leads to $\mu$P. The result is novel in two aspects:
>   1. We introduce new technical tools for analyzing neural network parametrization, going beyond LLN and CLT commonly used in literature.
>   2. The $\Theta\left(\frac{1}{\sqrt{d\log K}}\right)$ scaling rate is drastically different from existing results on all the other model component (e.g., $\Theta(m^{-1})$ for width scaling and $\Theta(L^{-1/2})$ for depth scaling). Directly applying existing result does not work for FNOs.
>
> * **On the algorithm side:** Based on our derived scaling rates in $\mu$P, we introduce $\mu$Transfer-FNO for zero-shot hyper-parameter transfer in FNO.
>
> * **On the experiment side:** We validate our theoretical claims on various PDEs, different training algorithms, and different hyper-parameters. The experiments consistently show the robustness of the theory. We also demonstrate that $\mu$Transfer-FNO can significantly reduce computational costs while maintaining or improving accuracy in practice.
>
> In particular, we believe that scaling up FNO is meaningful. Conceptually, FNO is designed to model continuous functions and enjoys the resolution-invariant property which standard Transformers do not have, making it a strong candidate for modeling PDE data. Practically, FNO is popularly used in recent research on pretraining and foundation models [1-4]. Therefore, we expect that our findings are relevant and interesting to the community.
>
> We also fully agree with the reviewer that Transformers are a powerful backbone for building foundation models. Developing techniques to scale up Transformers is also an important research direction, but it is orthogonal to the focus of this work. We appreciate your references and will add discussions on these related works in the revision.
>
> [1] Towards Foundation Models for Scientific Machine Learning: Characterizing Scaling and Transfer Behavior, NeurIPS 2023
>
> [2] Data-Efficient Operator Learning via Unsupervised Pretraining and In-Context Learning, NeurIPS 2024
>
> [3] Pretraining Codomain Attention Neural Operators for Solving Multiphysics PDEs, NeurIPS 2024
>
> [4] UPS: Efficiently Building Foundation Models for PDE Solving via Cross-Modal Adaptation, TMLR 2024/11
>
> **Regarding the dataset choice of our work.**
>
> We first point out that the primary focus of our experiments is to validate the robustness and generality of our theoretical claims, rather than benchmarking on the most challenging PDE datasets. The datasets we used are representative and commonly used in existing research.
>
> We also agree with the reviewer that mixed equation training is a common setting in PDE foundation model training. Following your suggestion, we conducted additional experiments on a mixed equation dataset involving Burgers' Equation, Advection Equation, and Reaction-Diffusion Equation using the data curated in [5]. Similar to the experimental setup in our submission, we first train FNOs with $K=6$ using different learning rates. Then train FNOs with $K=24$ under Standard Parametrization/$\mu$Transfer-FNO. We present the loss for each run below:
>
> | $\log_{10}$ (learning rate)       |  -2.0   |  -2.2   |  -2.4   |  -2.6   |  -2.8   |    -3.0     |    -3.2     |  -3.4   |  -3.6   |
> | --------------------------------- | :-----: | :-----: | :-----: | :-----: | :-----: | :---------: | :---------: | :-----: | :-----: |
> | $K=6$                             | 0.08314 | 0.04933 | 0.04665 | 0.03985 | 0.03822 | **0.03576** |   0.05366   | 0.05672 | 0.06318 |
> | $K=24$ (Standard Parametrization) | 0.98508 | 0.03785 | 0.03904 | 0.03200 | 0.03041 |   0.02852   | **0.02616** | 0.02989 | 0.03191 |
> | $K=24$ ($\mu$Transfer-FNO, ours)  | 0.03836 | 0.03517 | 0.03391 | 0.03192 | 0.02842 | **0.02599** |   0.02728   | 0.02925 | 0.03406 |
>
> The results are consistent with our original findings: On the mixed equation dataset, the optimal hyper-parameter shifts when the model size scales up under standard parametrization. In contrast, $\mu$Transfer-FNO stabilizes the optimal configuration with the lowest loss consistently obtained with the learning rate being $10^{-3.0}$, enabling zero-shot optimal hyper-parameter transfer from small models to large ones. We hope the additional result strengthen our work. Thank you for the constructive comment!
>
> [5] PDEBench: An Extensive Benchmark for Scientific Machine Learning, NeurIPS 2022
>
> We sincerely hope that our responses address your concerns and that you reevaluate our work based on the additional information. Thank you again for your time!

---

### Official Review · Reviewer_apVL · 2025-03-13

**Overall Recommendation:** 3

**Summary:**

This paper introduces μTransfer-FNO, a zero-shot hyperparameter transfer method for Fourier Neural Operators (FNOs). The core idea is to derive a Maximum Update Parametrization (μP) for FNOs that allows hyperparameters tuned on small FNOs to be directly transferred to larger FNOs without additional tuning, even for models with billions of parameters. The paper theoretically derives the μP for FNOs under the scaling of Fourier modes $K$, finding that initialization variances of kernel integral parameters should be scaled by $O\Big(\dfrac{1}{d \cdot \log K}\Big)$ and learning rates by $O\Big(\dfrac{1}{\sqrt{d \cdot \log K}}\Big)$, where $d$ is the PDE dimensionality. Experiments on Burgers' Equation, Darcy Flow, and the Navier-Stokes Equation demonstrate that μTransfer-FNO maintains optimal learning rates, batch sizes, and optimizer configurations across model scales, reducing computational tuning costs while preserving or improving accuracy. The method is also shown to be applicable to Physics-Informed Neural Operators (PINOs).

**Claims And Evidence:**

The paper supports its claims with evidence, but some areas could benefit from further clarification or stronger support:

**Strengths:**
* **µP Derivation:** The mathematical derivation of the µP for FNOs appears sound, although verifying the full proof requires careful examination of the appendix.
* **Learning Rate Stability:** The experiments clearly demonstrate that µTransfer-FNO stabilizes the optimal learning rate across different K values, supporting the central claim. Figures 1 and 2 provide visual evidence of this stability.
* **Generalizability to PINO:** The extension to Physics-Informed Neural Operators (PINOs) is well-supported by the experiments on Darcy flow (Figure 2c).
* **Transfer of Other Hyperparameters:** The experiments on batch size and Adam's $\beta_2$ (Figure 3) provide evidence that µTransfer-FNO can transfer hyperparameters beyond the learning rate, strengthening the overall argument.

**Weaknesses:**
* **Instability at Small $K$:** The paper acknowledges that the optimal learning rate can deviate at very small $K$ values, attributing it to training randomness. While plausible, further investigation or discussion of this instability would be beneficial. Perhaps providing error bars or statistics across multiple runs would strengthen this point.
* **Computational Cost Savings:** While Table 1 shows computational savings for the Navier-Stokes equation, the Darcy flow experiment shows increased training cost. The paper argues this is due to the smaller gap between small and large models in this setting. However, a more thorough analysis of the computational cost trade-offs, perhaps considering a wider range of model sizes, would be helpful. Quantifying the computational cost of the hyperparameter search itself would also be valuable.
* **Test Error Improvement:** The paper claims lower test error with µTransfer-FNO, attributing it to the ability to explore a larger hyperparameter search space with smaller models.While this is a reasonable explanation, it would be stronger to directly compare against a baseline that uses the same larger search space but tunes the large model directly (albeit at higher computational cost). This would isolate the benefit of µTransfer-FNO from simply using a larger search space.
* **Limited Scope of PDEs:** While the chosen PDEs are representative, exploring a wider range of PDE problems (PDEBench) would further strengthen the claim of generality. Including more complex or higher-dimensional PDEs would be particularly valuable.

**Essential References Not Discussed:**

The paper covers the most relevant literature regarding FNOs, PINNs, and µP/µTransfer. However, a few potential areas could benefit from additional discussion or referencing:

* **Theoretical Analysis of Hyperparameter Transfer:** While the paper provides a theoretical justification for µP in FNOs, it could benefit from discussing any existing theoretical work on hyperparameter transfer in general. Are there any theoretical guarantees or bounds on the effectiveness of transferring hyperparameters across different model sizes or architectures? Connecting to this broader theoretical literature would strengthen the paper's contribution.

* **Alternative Parametrizations:** The paper focuses on µP, but other parametrization schemes might exist for FNOs. Discussing potential alternatives and their potential advantages or disadvantages compared to µP would provide a more complete picture. For example, are there parametrizations specifically designed for different optimization algorithms or different types of PDEs?

**Experimental Designs Or Analyses:**

I've reviewed the experimental designs and analyses presented in the paper. While they provide support for the claims, there are some areas where the soundness and validity could be improved:

* **Hyperparameter Search Space:** The paper doesn't explicitly define the hyperparameter search space $\Xi$ used in Algorithm 1. Knowing the range and granularity of the search for each hyperparameter (learning rate, batch size, $\beta_2$) is crucial for interpreting the results. A larger search space could lead to better results regardless of µTransfer-FNO, so specifying the search space is essential for a fair comparison.

* **Baseline Comparison:** As mentioned previously, the comparison with the baseline of directly tuning the large model is not entirely fair. µTransfer-FNO benefits from exploring a potentially larger search space with the smaller proxy model. A stronger baseline would involve using the same expanded search space for tuning the large model directly, even if it's computationally more expensive. This would isolate the benefit of the transfer method itself.

* **Limited Number of Runs:** The paper presents results for single runs of each experiment. Given the stochastic nature of neural network training, reporting results averaged over multiple runs (e.g., $10-20$) with standard deviations or error bars would provide a more robust evaluation and account for potential variability.

* **Computational Cost Analysis:** The analysis of computational cost is somewhat limited. Table 1 only provides relative training costs ($1\times$, $1.38\times$, $0.30\times$). Reporting absolute training times or FLOPs would be more informative. Furthermore, the cost of the hyperparameter search itself is not factored into the comparison. A more comprehensive analysis should include the total cost of both the search and the final training.

* **Lack of Ablation Study:** An ablation study investigating the individual contributions of the initialization variance scaling and the learning rate scaling would provide further insights into the effectiveness of the proposed µP. This would help determine the relative importance of each component.

* **Details of Data Generation:** While the paper describes the PDEs used, more details about the data generation process would improve reproducibility.

* **Code Availability:** Providing the code used for the experiments would significantly enhance reproducibility and allow for independent verification of the results.

**Methods And Evaluation Criteria:**

Yes, the methods and evaluation criteria make sense for the problem of hyperparameter transfer in FNOs for PDE solving.
* **µP as a Method:** The use of µP as a foundation for hyperparameter transfer is well-motivated.The core idea of maintaining consistent training dynamics across model sizes is directly relevant to the goal of transferring hyperparameters. The theoretical derivation provides a principled basis for the proposed scaling factors.

* **Algorithm 1:** The µTransfer-FNO algorithm is a straightforward and logical application of the µP theory. It clearly outlines the steps involved in transferring hyperparameters from a small proxy model to a larger target model.

* **Choice of PDEs:** The selected PDEs (Burgers' Equation, Darcy Flow, and Navier-Stokes Equation) represent a reasonable range of complexity and dimensionality. They are commonly used as benchmarks in the FNO literature, allowing for comparison with existing work. While a broader set of PDEs (PDEBench) would be even better, the chosen set provides a good starting point.

* **Evaluation Metrics:** Using the L2 relative test error is a standard and appropriate metric for evaluating the performance of PDE solvers. It directly measures the accuracy of the solution obtained by the FNO.

* **Comparison with Baseline:** Comparing µTransfer-FNO against directly tuning the large model provides a relevant baseline.This allows for assessing the computational cost savings and potential performance gains of the proposed method. However, as mentioned in the previous response, a stronger baseline would involve using the same expanded hyperparameter search space for both methods.

**Other Comments Or Suggestions:**

* Page 5: "descritized" needs to be replaced by "discretized"
* Page 7: "serveral" needs to be replaced by "several"

**Other Strengths And Weaknesses:**

In summary, the paper is a valuable contribution, excelling in originality and practical use. Strengths and weaknesses were detailed in the preceding sections.

**Questions For Authors:**

Questions were depicted in the preceding sections.

**Relation To Broader Scientific Literature:**

This paper's main contributions relate to various areas within the broader scientific literature:

* **Fourier Neural Operators (FNOs) and Operator Learning:** The paper directly builds upon the existing literature on FNOs for solving PDEs (Li et al., 2021).

* **Physics-Informed Neural Networks (PINN):** The paper extends its method to Physics-Informed Neural Operators (PINOs) (Li et al., 2024), demonstrating its applicability beyond standard supervised learning for PDEs.

* **Maximal Update Parametrization (µP) and µTransfer:** The core theoretical contribution relies heavily on the µP framework introduced by Yang & Hu (2021) and the concept of µTransfer (Yang et al., 2022).

* **Hyperparameter Transfer Learning:** The overall goal of the paper is to enable efficient hyperparameter tuning for large models by transferring knowledge from smaller models.

**Theoretical Claims:**

I've reviewed the provided proof of Theorem 3.5, which establishes the µP for FNOs. While the overall structure of the proof seems reasonable, there are some specific points and potential issues:

* **Spectral Norm Calculation:** The proof relies heavily on calculating the spectral norm of the kernel integral operator $\mathcal{K}$. The argument that this norm is equivalent to the maximum absolute value of the parameters r seems plausible given the diagonal structure of $\mathcal{R}$ after the Fourier transform. However, the interaction between the multidimensional Fourier transform, the truncation operator $\mathcal{T}_K$, and the parameter tensor $\mathcal{R}$ could be complex, and a more detailed justification of this step would be beneficial.

* **High-Probability Bound:** The proof invokes a standard result from high-dimensional probability to bound the maximum of $K^d$ sub-Gaussian random variables. While this is a common technique, the specific constants and assumptions underlying this result need careful checking against the properties of the parameters $r$. The paper assumes these are sub-Gaussian, which needs to be verified in practice.

* **Simplification to $m=1$:** The proof simplifies the analysis by assuming a hidden dimension $m=1$. While extending to general $m$ is mentioned as straightforward, explicitly showing this extension, or at least providing a sketch, would strengthen the proof. The interaction between $m$ and $K$ in the scaling factors could be non-trivial.

* **Discretization Effects:** The proof works with the discretized version of the FNO. While this is necessary for practical implementation, the impact of discretization on the theoretical results is not explicitly discussed. Ideally, the proof should connect back to the continuous formulation of FNOs.

---

> ### Author Rebuttal · Authors · 2025-04-01
>
> Thank you for supporting our paper! We respond to your main questions and concerns below.
>
> **Regarding the scope of PDEs.** Following your and other reviewers' suggestions, we conduct additional experiments on a mixed equation dataset involving Burgers' Equation, Advection Equation, and Reaction-Diffusion Equation from PDEBench. This mixed equation dataset setting is relevant for PDE foundation model training as noted by other reviewers. We first train FNOs with $K=6$ using different learning rates, then train FNOs with $K=24$ under both Standard Parametrization and $\mu$Transfer-FNO. We present the loss for each run below:
>
> | $\log_{10}$(learning rate)        |  -2.0   |  -2.2   |  -2.4   |  -2.6   |  -2.8   |    -3.0     |    -3.2     |  -3.4   |  -3.6   |
> | --------------------------------- | :-----: | :-----: | :-----: | :-----: | :-----: | :---------: | :---------: | :-----: | :-----: |
> | $K=6$                             | 0.08314 | 0.04933 | 0.04665 | 0.03985 | 0.03822 | **0.03576** |   0.05366   | 0.05672 | 0.06318 |
> | $K=24$ (Standard Parametrization) | 0.98508 | 0.03785 | 0.03904 | 0.03200 | 0.03041 |   0.02852   | **0.02616** | 0.02989 | 0.03191 |
> | $K=24$ ($\mu$Transfer-FNO)        | 0.03836 | 0.03517 | 0.03391 | 0.03192 | 0.02842 | **0.02599** |   0.02728   | 0.02925 | 0.03406 |
>
> The results are consistent with our original findings: On the mixed equation dataset, the optimal hyper-parameter shifts when the model size scales up under standard parametrization. In contrast, $\mu$Transfer-FNO stabilizes the optimal configuration. We believe these additional results strengthen our work. Thank you for the constructive comment!
>
> **Regarding our theoretical results.**
>
> - **Spectral Norm Calculation:** Our proof is applicable to a general dimension $d$. We only use the fact that the Fourier transform is orthogonal, and that the Fourier transform, the truncation operator, the multiplication with the parameter tensor, and the inverse Fourier transform are all linear operators. All these properties hold regardless of the dimensionality.
>
>   **The sub-Gaussian assumption:** We note that our analysis focuses on the Adam optimizer which uses entry-wise normalized gradient momentum for updates. Furthermore, one can optionally apply value-based gradient clipping in practice, which strictly enforces all entries of the update to be bounded and hence sub-Gaussian. We have empirically verified this assumption in our preliminary experiments and will add a remark on this in the paper revision.
>
>   **Extending to the case with a general $m$:** In this setting, the analysis can be broken down by dealing with each entry of the matrix-vector product separately and writing the product as the summation of entry-wise multiplication. Then one can treat $\widetilde{\boldsymbol R}_{\ell}\in\mathbb{R}^{N_1 \cdots N_d\times N_1 \cdots N_d\times m\times m}$ as $m\times m$ instantiations of $N_1 \cdots N_d\times N_1 \cdots N_d$ matrices and apply the analysis in the $m=1$ case to arrive at the same result. Based on this argument, a general $m$ would not incur complicated dependency on the scaling rate of $K$.
>
>   **Discretization Effects:** Our proof does not rely on any property of a specific resolution. We mainly leverage the fact that the Fourier transform matrix is an orthogonal matrix in the discrete formulation, and that the main building blocks in the kernel integral operators are all linear. Connecting to the continuous formulation of FNOs, these facts can be understood as consequences of orthogonality and linearity of the corresponding operators over functional spaces.
>
> **Regarding theoretical analysis on hyper-parameter transfer.** To the best of our knowledge, no theoretical bounds exist for hyper-parameter transfer across different sizes or architectures. Most existing work, including our results, focuses on asymptotic scaling rates, building on or adjacent to the $\mu$Transfer framework as discussed in Section 5.3. Another relevant recent result is [1], which we will include in our discussion.
>
> [1] Super Consistency of Neural Network Landscapes and Learning Rate Transfer, NeurIPS 2024
>
> **Regarding alternative parametrizations.** This is an interesting question! Our analysis is not tied to any specific PDE, similar to how other existing analyses are not tied to any specific task or dataset. However, $\mu$Parametrization does vary for different optimization algorithms. For example, [2] derives $\mu$P for K-FAC and Shampoo optimizers. Our analysis focuses on Adam because of its popularity.
>
> [2] On the parameterization of second-order optimization effective towards the infinite width.
>
> Due to the word limit, we cannot respond to other comments individually. But we would like to assure you that we value your comments and will modify our paper accordingly based on your comments. We sincerely hope that our responses address your concerns!

---

### Official Review · Reviewer_Xe5F · 2025-03-13

**Overall Recommendation:** 4

**Summary:**

The authors discuss how Fourier Neural Operators (FNOs), which is a state-of-the-art SciML method, have been used to solve complex PDEs. However, they identify issues with scaling FNO to more intricate PDEs that requires increasing the number of Fourier modes. Increasing the number of Fourier modes increases the number of model parameters and makes HPO very expensive. The authors propose $\mu$Transfer-FNO, which is a zero-shot hyperparemeter transfer method with optimal hyperparameters tunes on smaller FNOs to be applied to large billion-parameter FNOs zero-shot. The method is based upon the Maximum Update Parametrization ($\mu$P) framework. the authors show that $\mu$Transfer-FNO reduces the cost of tuning parameters on large FNOs and maintains the accuracy.

## Update after rebuttal
I appreciate the authors' detailed rebuttal, motivating example and additional experiments. In light of that, I raised my score.

**Claims And Evidence:**

- The proposed transfer method makes sense to help scale HPO tuning for large FNOs. I would like to see a concrete example where larger Fourier modes are required to solve the PDE for better problem motivation.
- The authors correctly motivate that FNO scales as $\mathcal{O}(K)^d$, where $K$ is the number of Fourier modes and $d$ is the dimension of the problem, solving that the complexity grows exponentially with the dimension so for practical 3D space and time problems, this can be very expensive.
- Rather than showing the loss in Figure 1 , it may be more informative to show the validation accuracy as a more meaningful metric. It is interesting that the optimal learning rate with the proposed method remains approximately the same regardless of the model size. To better show this, it may be good to put a vertical line through the star points.

**Essential References Not Discussed:**

- It is good that the authors include the known boundary conditions in the problem definition in Eqn. 1. There is missing reference to Saad et al., "Guiding continuous operator learning through Physics-based boundary constraints", ICLR 2023 that enforces boundary conditions as an exact constraint.
- The authors mention that FNO is most commonly NO. They should state the reason is because the FFT is very computational efficient to compute the kernel vector $\mathcal{K}x$ products. Having said that, the paper is missing references to other neural operators with different bases, e.g.,  Gupta et al., “Multiwavelet-based Operator Learning for Differential Equations”. In: Advances in Neural Information Processing Systems. Vol. 34. PMLR, pp. 24048–24062, 2021, Li, Zongyi, Nikola Kovachki, et al. (2020b). “Multipole Graph Neural Operator for Parametric Partial Differential Equations”. In: arXiv preprint arXiv:2006.09535.
- It is good that the authors also compare to PINO. I think the authors should be a bit careful referring to the PINNs loss as "more advanced training techniques". Both the PINO paper and Saad et al., ICLR 2023 show that in several cases the unconstrained FNO outperforms PINO. Also there have been several instabilities with PINNs training that have been reported in the literature that are not cited, e.g., Krishnapriyan et al., "Characterizing possible failure modes in physics-informed neural networks", Advances in neural information processing systems 34, 26548-26560, 2021. This looks like it is also observed by the authors own training of PINO in Figure 2(c) with the spike and oscillations in the loss function.
- References to transfer learning methods other than $\mu$ P (Yang & Hu (2021) and $\mu$Transfer methods (Yang et al. (2022)) are missing.

**Experimental Designs Or Analyses:**

- Nice that experiment section is organized according to various questions to show that aforementioned theory also holds empirically.
- Please clarify that Burgers' in 4.1 is actually viscous Burgers' for $\nu \ne 0$, which is simpler to solve than the sharp shock solution case where $\nu = 0$ and there is no artificial diffusion from the viscosity term.
- Good that the authors consider 1D and 2D problems and also vary mapping the initial condition to solution in Burgers' and PDE coefficient to the solution in Darcy Flow.
- Since one of the main motivations of the proposed method is for large-scale problems, I think 3D spatial cases should also be used because that when $d=3$ is when the FNO becomes very computational expensive. It is good that 3D Navier-Stokes is tested. Is this FNO-3D because it is the standard 2D space + time test case? I think 3D space should also be considered. An example practical, real-world 3D Car Surface Pressure Prediction test case is provided in Ma et al., "Calibrated uncertainty quantification for operator learning via conformal prediction", 2024 and Li et al., " Geometry-informed neural operator for large-scale 3d pdes", 2023. Both of these works should be cited as well.

**Methods And Evaluation Criteria:**

- It makes sense to try to increase the number of Fourier modes and the proposed transfer HPO method seems effective. To do so, the proposed method scales the kernel integral operator parameters as the number of modes $K$ is increased so that the optimal hyperparameters stay approximately the same across model sizes. This motivates HPO on the small model and then directly transferring to larger FNOs.
- The result cover a good range of PDEs, i.e., Burgers, Darcy Flow and the challenging Navier-Stokes equations. I would like to see a larger number of equations tested. See the comprehensive Neural Operator benchmark in Saad et al., "Guiding continuous operator learning through Physics-based boundary constraints", ICLR 2023.
- The authors show that the proposed transfer learning preserves the same optimal learning rate, training batch size and optimizer configurations across increasing model sizes.
- With the proposed approach, the method is able to scale to large FNOs with nearly 1B parameters with better accuracy and only `0.3x` training compute.
- In Figure 2, even with the proposed approach the optimal learning rates are not exactly the same and seem slightly shifted but less than the base especially in Figure 2a-b. Is there a quantitative metric to better measure this difference?

**Other Comments Or Suggestions:**

Typos
- Line 221 left column, \citet should be used instead of \citep for the reference.
- Capital W on line 374 right column to start the sentence.
- "serveral" on line 413 left.

**Other Strengths And Weaknesses:**

## Weaknesses
- The author should better define what they mean by "intricate PDEs" in the abstract.
- The introduction is missing a contribution section. I would also move Theorem 1.1 to the method section and clearly state the contributions in the Introduction.
- The authors should define $a_i$ in the training pairs, e.g., initial conditions, PDE parameters. Using $a_i$ for the notation is confusing since later in 4.1 a is used as the Darcy coefficient as well and sometimes for Burgers the initial condition$u_0$ is used in the mapping


## Strengths
- The authors address a common problem with FNOs in scaling to higher dimensions, i.e., the curse of dimensionality and propose a simple transfer learning approach of the hyperparameters to scale to larger FNOs with more modes to allow costly HPO on these larger architectures.

**Questions For Authors:**

1. How would this method apply to more general NOs than just FNO, e.g., the Multi-wavelet NO in Gupta et al., 2021?
2. Are any of the NO method exampels in this work including time or extrapolation in time like the Markov Neural Operator, Li et al., "Learning Dissipative Dynamics in Chaotic Systems", 2021.
3. Why is the total training cost increased from tuning the full model for Darcy Flow in FNO-2D but decreased for FNO-3D in Table 1. Also the table caption states that three PDEs are shown in the table but it only shows 2.

**Relation To Broader Scientific Literature:**

- FNOs have shown large impact on solving PDE problems with ML. This paper focuses on more challenging PDEs where larger parameter FNOs are needed with more Fourier modes. The authors in the introduction go directly into discussing the specifics of FNO and the effect of its modes. I think the authors should first motivate solving PDE problems, why ML methods have been developed to solve these problems, mention the three main method classes, e.g., PINNs, Neural Operators and MeshGraphNets and then state that this paper will focus on FNOs. The authors can tell clearly identify the more challenging PDEs with regular size parameters FNOs struggle to motivate these larger-parameter FNO models which require HPO.
- The authors mention the $\mu$ P (Yang & Hu (2021) and $\mu$Transfer methods (Yang et al. (2022)). Please clearly differentiate the novelty of the proposed approach. Does $\mu$Transfer-FNO just directly apply these methods to FNO? If so, please explain the technical challenges of applying them to the FNO architecture.
- My main concern is that the proposed method is too similar to the past Yang & Hu (2021)  and Yang et al. (2022) in literature and hence the novelty needs to be clarified.
- The authors state that the number of hidden dimensions $m$ and model depth $L$ are fixed in this work and have been studied in  Yang & Hu (2021) and Yang et al. (2022). Please briefly summarize their effect here or in an appendix so that this work can be fully understood end-to-end without relying on these prior works for key concepts.

**Theoretical Claims:**

- The main theoretical result is presented in Theorem 3.5  on calculating the scaling rate for kernel up the FNO kernel integral operator. The theorem shows the abc-parametrization is a $\mu$P of FNO with the Adam optimizer and scales according to 1 / sqrt(d), where d is the dimensionality. The proof is detailed and provided in the appendix with a proof outline of the main concepts in the main body.
- The authors highlight a difference in their proof than in the standard $\mu$P proofs that the past proofs rely on the CLT to measure the average of random variables, whereas there is a technically difference in the structure of FNO and the kernel integral operator that leads to the analysis of the maximum of the random variables. It is good that the authors clarify this difference.
- It is also good that the proposed scaling function in the proof is not resolution dependent so that the desired resolution invariance property of NOs is not broken with this methods - do the authors have proof of this, i.e.,that resolution invariance still holds?

---

> ### Author Rebuttal · Authors · 2025-04-01
>
> Thank you for supporting our paper! We respond to your main questions and concerns below.
>
> **Regarding motivating the need for larger Fourier modes.** A concrete example is based on Kolmogorov microscales in fluid dynamics: When simulating turbulent flows governed by the Navier-Stokes equations, the Kolmogorov microscales represent the smallest scales at which energy dissipation occurs, which are proportional to $\mathrm{Re}^{-3/4}$ where $\mathrm{Re}$ is the Reynolds number. For realistic engineering applications with high Reynolds numbers, these microscales become extremely small and require careful modeling of high-frequency features. In such cases, larger numbers of Fourier modes are preferred.
>
> **Regarding more PDEs.** Following your and other reviewers' suggestions, we conduct additional experiments on a mixed equation dataset involving Burgers' Equation, Advection Equation, and Reaction-Diffusion Equation from PDEBench. This mixed equation dataset setting is relevant for PDE foundation model training as noted by other reviewers. We first train FNOs with $K=6$ using different learning rates, then train FNOs with $K=24$ under both Standard Parametrization and $\mu$Transfer-FNO. We present the loss for each run below:
>
> | $\log_{10}$(learning rate)        |  -2.0   |  -2.2   |  -2.4   |  -2.6   |  -2.8   |    -3.0     |    -3.2     |  -3.4   |  -3.6   |
> | --------------------------------- | :-----: | :-----: | :-----: | :-----: | :-----: | :---------: | :---------: | :-----: | :-----: |
> | $K=6$                             | 0.08314 | 0.04933 | 0.04665 | 0.03985 | 0.03822 | **0.03576** |   0.05366   | 0.05672 | 0.06318 |
> | $K=24$ (Standard Parametrization) | 0.98508 | 0.03785 | 0.03904 | 0.03200 | 0.03041 |   0.02852   | **0.02616** | 0.02989 | 0.03191 |
> | $K=24$ ($\mu$Transfer-FNO)        | 0.03836 | 0.03517 | 0.03391 | 0.03192 | 0.02842 | **0.02599** |   0.02728   | 0.02925 | 0.03406 |
>
> The results are consistent with our original findings: On the mixed equation dataset, the optimal hyper-parameter shifts when the model size scales up under standard parametrization. In contrast, $\mu$Transfer-FNO stabilizes the optimal configuration. We believe these additional results strengthen our work. Thank you for the constructive comment!
>
> **Regarding the novelty compared to existing works on $\mu$P and $\mu$Transfer.** While the high-level algorithmic idea of $\mu$Transfer-FNO is based on existing research, we are the first to derive the Maximum Update Parametrization ($\mu$P) for FNOs. The result is novel in two aspects:
>
> 1. We introduce new technical tools for analyzing neural network parametrization, going beyond LLN and CLT commonly used in literature.
>
> 2. The $\Theta\left(\frac{1}{\sqrt{d\log K}}\right)$ scaling rate is drastically different from existing results on all the other model components (e.g., $\Theta(m^{-1})$ for width scaling and $\Theta(L^{-1/2})$ for depth scaling).
>
> We point out that directly applying existing results does not work for FNOs because the design of the kernel integral operator is significantly different from other standard neural network modules such as embedding layers and linear transforms.
>
> **Regarding more general NOs than just FNO.** Unfortunately, our current theoretical analysis is specific to the kernel integral operator and FNO. We will mention this as a limitation of our work in the paper revision and include references to other NO variants including Multi-wavelet NO, Multipole Graph Neural Operator, etc.
>
> **Regarding extrapolation in time.** Our experiments focus on more standard settings for FNO and do not consider extrapolation in time like the Markov Neural Operator. We believe that our technique is still applicable to this setting since the design of FNO is unchanged.
>
> **Regarding Table 1.** Thank you for catching the typo! As for the total training cost, this difference occurs because the size of FNO scales as $\mathcal{O}(K)^d$. Thus, when $d$ increases, the efficiency gain from tuning models with small $K$ and applying $\mu$Transfer-FNO becomes much more significant. Specifically, the Darcy Flow problem with FNO-2D is relatively simple. Direct tuning on the full-sized model with a smaller hyper-parameter search space can still lead to decent results with reasonable computational cost. Hence, this setting is not particularly favorable to $\mu$Transfer-FNO. However, for the Navier-Stokes Equation with FNO-3D, tuning the full model becomes prohibitively expensive. In this more complex setting, $\mu$Transfer-FNO offers substantial efficiency gains by allowing us to tune models with small $K$ and then transfer the optimal hyper-parameters to larger models.
>
> Due to the word limit, we cannot respond to other comments individually. But we would like to assure you that we value your comments and will modify our paper accordingly based on your writing suggestions and references. We sincerely hope that our responses address your concerns!

---

### Official Review · Reviewer_tRrn · 2025-03-16

**Overall Recommendation:** 4

**Summary:**

This paper introduces $\mu$Transfer-FNO, a zero-shot hyperparameter transfer technique for Fourier Neural Operators (FNOs). Based on the Maximum Update Parametrization (μP) framework, the authors propose a parametrization scheme that enables hyperparameters tuned on smaller FNOs to be transferred directly to much larger models without retraining. The authors derived a novel scaling law for the parameters of FNOs with respect to the number of Fourier modes. Extensive experiments on multiple PDEs like the Burgers' equation, Darcy Flow equation and the Navier-Stokes equation together demonstrate that μTransfer-FNO enables scaling of the nearly billion-parameter FNOs while reducing computational cost and maintaining accuracy.

## Update After Rebuttal
The reviewer is positive about the results presented in the paper and satisfied with the authors' responses to the questions raised in all reviews. Hence, the reviewer would like to remain the score.

**Claims And Evidence:**

Here is the main claim made in this paper: $\mu$Transfer-FNO allows zero-shot hyperparameter transfer from small-scale FNOs to large-scale FNOs. Sufficiently many experiments are included to compare $\mu$Transfer-FNO with standard parametrization, which supports the claim in a clear and convincing way.

**Essential References Not Discussed:**

Below is one preprint that probably needs to be discussed in the paper is [1]. Specifically, the main result in [1] discussed the connection between neural operators and transformers (the attention architecture), which is also partially addressed in the cited paper [2]. The authors should discuss how the $\mu$Transfer-FNO methodology proposed in this paper differs from the $\mu$Transfer framework originally proposed for transformers, since neural operators can be linked with transformers under certain circumstances.

References:

[1] Calvello, Edoardo, Nikola B. Kovachki, Matthew E. Levine, and Andrew M. Stuart. "Continuum attention for neural operators." arXiv preprint arXiv:2406.06486 (2024).

[2] Kovachki, Nikola, Zongyi Li, Burigede Liu, Kamyar Azizzadenesheli, Kaushik Bhattacharya, Andrew Stuart, and Anima Anandkumar. "Neural operator: Learning maps between function spaces with applications to pdes." Journal of Machine Learning Research 24, no. 89 (2023): 1-97.

**Experimental Designs Or Analyses:**

Yes, please refer to the "Methods And Evaluation Criteria" section above.

**Methods And Evaluation Criteria:**

Yes. The experiments are mainly based on the Burgers’ Equation (1D), the Darcy Flow Equation (2D), and the Navier-Stokes Equations (3D), which are standard examples used in the original FNO paper. The authors studies the performance of $\mu$Transfer-FNO on the finetuning of multiple parameters like the learning rates and batch sizes with respect to different number of Fourier modes. Experiments on other variants of FNO like PINO (Physics-Informed Neural Operator) are also included to justify the efficacy of the proposed methodology.

**Other Comments Or Suggestions:**

N/A

**Other Strengths And Weaknesses:**

To the best of the reviewer's knowledge, this is one of the first work that studies how to finetune parameters for large-scale FNOs based on good parameters trained on small-scale FNOs, which is of not only practical significance but also theoretical insights. For future work, it might be meaningful to study the proposed methodology under the setting of infinitely wide neural networks proposed in the series of paper on tensor programs. However, one potential weakness of this work is that it might be meaningful to test the proposed methodology on other architectures within the operator learning framework, such as DeepONets.

**Questions For Authors:**

N/A

**Relation To Broader Scientific Literature:**

This work is based on the Maximal Update Parametrization ($\mu$P) framework proposed in the series of work on tensor programs, which is originally used in the studies of MLPs and Transformers. However, this paper should be mainly posited within the literature (FNOs, DeepOnets and their variants) on operator learning and neural PDE solvers.

**Theoretical Claims:**

The only theoretical claim in the paper is Theorem 1.1 (informal version of Theorem 3.5). Its proof in the supplement has been verified to be correct.

---

> ### Author Rebuttal · Authors · 2025-04-01
>
> Thank you for supporting our paper! We respond to your questions and concerns below.
>
> **Regarding the connection to Transformers.** This is an interesting point! The Fourier Integral Operator and Continuum Attention are both nonlocal operator classes, but they have different parametrizations:
>
> - For Continuum Attention, the parametrization closely resembles vanilla attention, and its size is controlled by the hidden dimensionality (the model "width"). Therefore, existing results on $\mu$Transfer for scaling up vanilla attention are directly applicable.
> - In contrast, the Fourier Integral Operator is parametrized in a uniquely different way—by modeling the Fourier Transform of a kernel function with a pre-defined number of Fourier modes (i.e., $K$ in our paper's notation). This design differs from all modules studied in existing Maximum Update Parametrization ($\mu$P) and $\mu$Transfer literature, and our findings show that its scaling rate is also drastically different from that of other model components.
>
> That being said, in Fourier Integral Operators, the notion of model "width" still exists (corresponding to $m$ in our paper's notation). The size of Fourier Integral Operators is controlled by both $m$ and $K$. Existing width scaling results apply to $m$, while our analysis applies to $K$, which is the unique aspect of Fourier Integral Operators.
>
> **Regarding to other architectures within the operator learning framework.** We acknowledge that our current findings are limited to FNOs. We believe it would be a meaningful future direction to study $\mu$P and $\mu$Transfer for other operator learning models, e.g., DeepONets.
>
> We sincerely hope that our responses address your concerns. We are also happy to discuss further if you have any additional questions. Thank you again for your time!

---

> > ### Comment · Reviewer_tRrn · 2025-04-06
> >
> > The reviewer would like to thank the authors for their clarification. The authors are encouraged to include a discussion on the relation between transformers (or the continuum attention architecture) and FNO, which might inspire future studies on how the $\mu$Transfer methodology differs for these two models. Overall, the reviewer remains positive about the results of this paper and would like to keep the score.

---

> > > ### Author Response · Authors · 2025-04-07
> > >
> > > We appreciate your continued support of our paper! We will follow your suggestion to include a discussion on the connections and differences between Transformers (particularly the continuum attention) and FNOs under the $\mu$P and $\mu$Transfer framework in Section 3.
> > >
> > > Thank you again for your thoughtful feedback that has helped strengthen our work.

---

### Decision · Program_Chairs · 2025-05-01

**Decision:**

Accept (poster)

**Comment:**

The paper introduces a hyperparameter transfer technique for FNOs based on the maximum update parameterization framework. The authors provide both theoretical and experimental evidence that their framework allows to transfer hyperparameters from small to large models. The reviewers raised concerns about context, evaluation, and framing, which the authors carefully addressed in their rebuttal, leading to one of the scores being raised.

The scores are 2 "Accept", "Weak accept", and "Weak reject". The "weak reject" review is primarily concerned with significance; specifically, the reviewer doesn't see the importance of studying and scaling FNOs (favoring transformers instead). This opinion is not universal, as evidenced by the other three reviews. The authors' rebuttal also carefully explains the significance and ubiquity of FNOs in the field.